# Influence of Leaching Solution on the Soil-Water Characteristics of Ion-Absorbed Rare Earth Minerals and Its Hysteresis Effect

**Zhong-qun Guo [1,2,*], Ling-feng Liu [1], Tao Tang [1], Ke-fan Zhou [1] and Xiao-jun Wang [2]**

[1] School of Civil Engineering and Surveying & Mapping Engineering, Jiangxi University of Science and Technology, Ganzhou 341000, China

[2] JiangXi Key Laboratory of Mining Engineering, Jiangxi University of Science and Technology, Ganzhou 341000, China

\* Correspondence: guozhongqun_jxust@163.com

**Abstract:** Thesoil-water characteristic curve is the basic constitutive relation to express the water-holding characteristics of ion-absorbed rare earth. The simulated solution mining test and pressure plate apparatus tests were carried out, the relationship between matric suction and water content under different leaching actions was obtained. Using the Fredlundand Xing four-parameter model, the soil-water characteristic curves of ion-absorbed rare earth under different leaching action-swereobtained, and the changing trend of each parameterwas analyzed, and the influence of different leaching methods on the water-holding characteristic of soil was obtained.For different types of leaching solutions, the water-holding capacity of the soil varies from strong to weak forpure water, the 3% magnesium sulfate solution, and the 3% ammonium sulfate solution; as the concentration of the leaching solution increases, the water-holding capacity of the sample gradually decreases, and it decreases most significantly from 0% to 2%. Moreover, the saliency of the "hysteresis effect" of the soil-water characteristic curve was from high to low forpure water, the 3% magnesium sulfate solution, and the 3% ammonium sulfate solution, and the "hysteresis effect" showed a decreasing trendwith the increaseinthe concentration.

**Keywords:** ion-absorbed rare earth; in situ leaching; soil–water characteristic curve; water-holding characteristics; leach solution

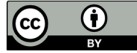

## 1. Introduction

Rare earths are indispensable raw materials in high-precision fields, such as many modern industries and the defense industry. Many of the world's rare earth products come, predominantly, from ion-absorbed rare earth ores (also referred to as weathered crust elution-deposited rare earth ores) in eight provinces of southern China [1, 2]. In theseores, rare earth elements, which are adsorbed intothe clay minerals through hydrated cations or hydroxyl-hydrated cations, are difficult to be enriched by conventional selection techniques [3–5]. Currently, in situ leaching technologyis widely recommended for the mining of ion-absorbed rare earth ores, which providesthe advantages of high efficiency, being environmentally friendly, and low-cost.In situ leaching iswidely used in areas with good geological conditions [6–8]; however, it is easy to cause landslides in mining areas throughimproper liquid injection in leaching, a process which is, also, especially affected by the abundant rainfall in southern China, and there are many other hidden dangers in the mining process [9–11]. The water-holding characteristics reflect the difficulty of pore water-change in the soil, which is the basis for studying the seepage law and strength characteristics of unsaturated soil, and there is an important relationship between the soil-water characteristics and landslides caused by oversaturation. Research on the

effects of differentleaching conditions on the water-holding characteristics of ion-absorbed rare earth orescan provide a theoretical basis for the safety of mining.

The soil-water characteristic curve is a graphical relation curve between matrix suction and water content, and it is relevant to the water-holding characteristics and shear strength of soil [12, 13]. Some scholars have carried out significant amount meaningful scientific research on this topic. Miller et al. [14] studied the variation of water content and pore water pressure in clay soil, and revealed the characteristics of unsaturated soil under different water content conditions. Zhou et al. [15] studied the effect of dry density on SWCC of unsaturated soil based on the incremental relationship between saturation and initial pore ratio. Miao et al. [16] analyzed the relationship between SWCC characteristics and pore structure. Vanapalli et al. [17] experimentally studied the influence of stress history on the water-holding characteristics of soil. Charles et al. [18] studied the effects of stress state, initial dry density, and initial water content on SWCC. Salager et al. [19] studied the influence of temperature on the water-holding characteristics of soil from both theoretical and experimental aspects, and established a SWCC theoretical model for predicting arbitrary temperatures. At present, the effects of different particle size, pore ratio and other factors on the soil-water characteristic curve have been studied for the soil-water characteristics of ion-absorbed rare earth [14, 15].

In regards to research related to engineering considerations of ion-absorbed rare earth, there are few studies concerning the effects of different types and concentrations of leaching solution on the soil-water characteristic curve.In this study, rare earth ore samples were collected from the Ganzhou Longnanin the Jiangxi Province of China.The drying and wetting curves of ion-absorbed rare earth under different leaching solutions were obtained through the Geo-Experts pressure plate apparatus, and a typical soil-water characteristic curve model wasused to analyze the influence of different leaching actions on the characteristic parameters of soil-water characteristic curves.The influence of different leaching actions on the water-holding characteristics and "hysteresis effect" of ion-absorbed rare earth were revealed.

## 2. Test Materials and Methods

### 2.1. Samples

Ion-absorbed rare earth ore samples are from Ganzhou Longnan Rare Earth Ore Mine in Jiangxi Province, China.The ore samples were taken from a depth of 0.5~1 m in the ore body. The basic physical parameters of the sample are listed in Table 1, the particle size distribution curve of the rare earth ore is shown in Figure 1.

**Table 1.** Basic physical parameters of rare earth ores.

| Physical Indexes | Water Content (%) | Density, $\rho$/(g/cm³) | Dry Density (g/cm³) | Liquid Limit, $W_P$/(%) | Plastic Limit, $W_P$/(%) | Plastic Index, $I_P$ |
|---|---|---|---|---|---|---|
| Original rare earth | 16.95 | 1.67 | 1.428 | 40.56 | 30.27 | 10.29 |

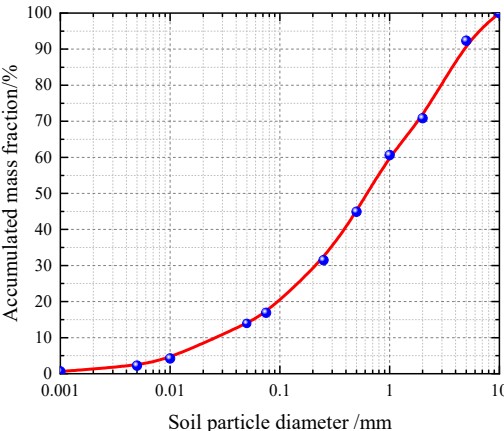

**Figure 1.** Particle size distribution curve of rare earth ores.

According to the dry density and water content of the undisturbed soil, the samples were remolded, sealed with plastic film for 24 h, and the cutting-ring was loaded into layers to prepare for the simulated leaching test.

### 2.2. Test Method

Based on the self-made leaching device, the simulation tests were carried out to study the effects of different types and concentrations on the soil-water characteristic curve of ion-absorbed rare earth, the leaching time was set to 48 h, the liquid–solid ratio was set to 0.6:1, and the leaching temperature was set to 25°. The test conditions were set as follows:

(1) Type of leaching solution: pure water, a 3% ammonium sulfate solution, and a 3% magnesium sulfate solution were selected for the leaching solution.

(2) Concentration of leaching solution: ammonium sulfate solution with concentrations of 0%, 2%, 3%, and 5% was selected for the leaching solution.

The sample after leaching were treated with24 h of vacuum saturation, and then put into the Geo-Experts pressure plate apparatus (Figure 2) for testing, the drying experiment was carried out under 0, 10, 20, 50, 100, 150, 200, 300, 400, and 480 kPa, after completing the drying process, then the wetting process is carried out in reverse. Equilibrium standards of samples under single-level matric suction were referred to suggestions of HQ Pham [20, 21], when the water discharge amount in 24 h is smaller than 0.1 mL, the matrix suction is considered to be stable.

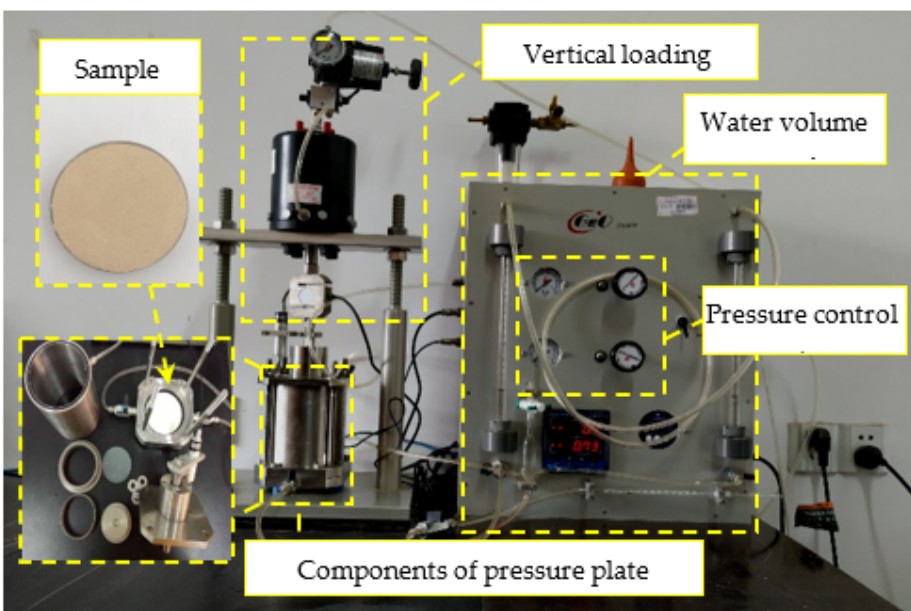

**Figure 2.** Geo-Experts pressure plate apparatus.

*2.3. Soil-Water Characteristic Curve Model*

Classical soil-water characteristic curve models include the Van Genuchten model [22], the FredlundandXing three-parameter model [23], and the FredlundandXing four-parameter model [24]. According to the conclusion of previous research, the FredlundandXing four-parameter model has good applicability for ion-absorbed rare earth research, so the FredlundandXing four-parameter model was selected, and its function expression is as follows,

$$\frac{\theta - \theta_r}{\theta_s - \theta_r} = \frac{1}{\{\ln[e + (\frac{\psi}{a})^n]\}^m} \tag{1}$$

where $\theta$ is the volume water content of the soil; $\theta_r$ is the residual water content, $\theta_s$ is the saturated water content; $\psi$ is the matrix suction; $a$, $n$, and $m$ are the model optimization parameters, where parameter $a$ is related to the air-entry value, and $n$ is a parameter related with drying rate and it controls slope of the soil-water characteristic curve; $m$ is a parameter related with residual water content and it is correlated with the overall symmetry of curve.Due to the air-entry pressure value of the clay plate is limited (5 bar clay plate is used in this test), the residual water content state in the test is not reached, so $\theta_r$ is obtained by data fitting.

## 3. Effect of Leaching Solution on Water-Holding Characteristics of Ion-Absorbed Rare Earth

*3.1. Effect of Leaching Solution Type on Water-Holding Characteristics*

The FredlundandXing four-parameter model was used to analyze the water-holding characteristics of ion-absorbed rare earth after leaching of different types of leaching solution, the horizontal  4xisdenotes the matrix suction$\psi$, the value range is 0.1~$10^6$ kPa, the vertical4xis denotes the volume water content$\theta$, and the soil-water characteristic curves of ion-absorbed rare earth after the leaching of different types of leaching solution are shown in Figure 3.

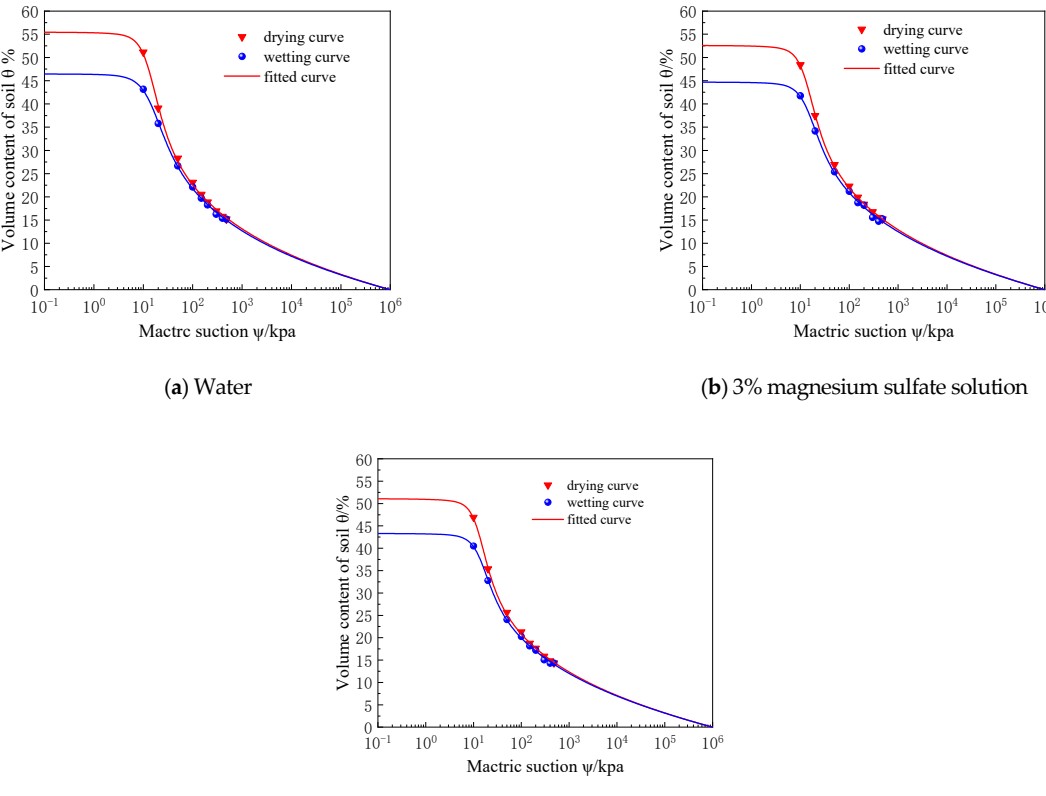

(**a**) Water

(**b**) 3% magnesium sulfate solution

(**c**) 3% ammonium sulfate solution

**Figure 3.** Soil-water characteristic curve under different types of leaching solution.

It can be seen that, during the drying process, with the increase inmatrix suction, the volume water content of the sample first decreases slowly, and then rapidly decreases until the matrix suction reaches $10^6$ kPa and the volume water content tends to 0. In the process of wetting, with the decrease inmatrix suction, thewater content increases in the opposite direction, but it is always less than the water content of the drying curve, and there is an obvious "hysteresis effect" between the two; the smaller the matrix suction, the more obvious the hysteresis effect.

In order to better compare the effects of different types of leaching solution, the drying and wetting curves of ion-absorbed rare earth after the leaching of different types of leaching solution are plotted, as shown in Figure 4. The analysis showed that the saturated water content of the sample gradually decreased after the leaching of pure water, the 3% magnesium sulfate solution, and the 3% ammonium sulfate solution, and the drying and wetting curve were gradually shifted downward. When the matrix suction is constant, the volumetric water content of the sample shows the law of pure water>3% magnesium sulfate solution>3% ammonium sulfate solution; that is, the water-holding capacity of the rare earth sample gradually weakens after the leaching of three types of leaching solution. As the matrix suction gradually increased to $10^6$ kPa, the drying and wetting curve after the leaching of the three types of leaching solution cooperated more clearly.

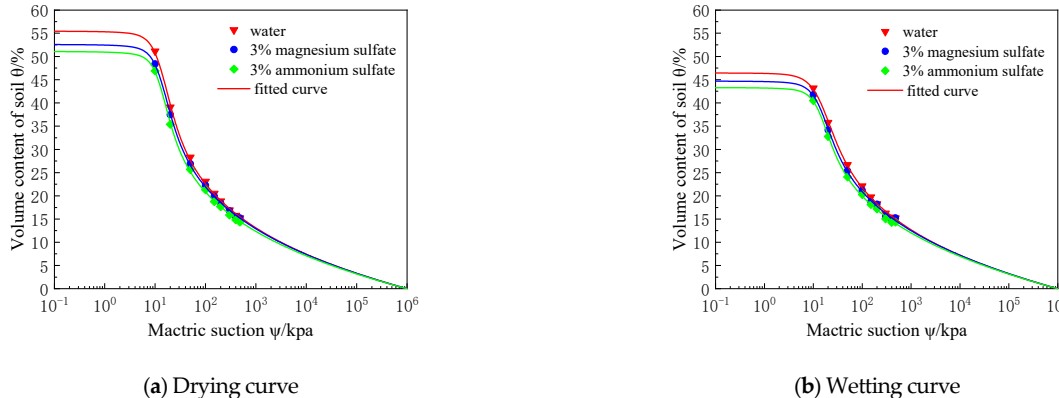

(**a**) Drying curve                    (**b**) Wetting curve

**Figure 4.** Drying and wetting curve under different types of leaching solution.

Rare earth ions in ion-absorbed rare earth ores are mainly adsorbed in the form of hydrated cations or hydroxyl-hydrated cations in clay minerals, such as kaolinite, mica, and feldspar in ores [25]. When ion-absorbed rare earth ores are leached with different leaching solutions, ion exchange will occur when the rare earth ions in the clay minerals encounter more active cations (egMg$^{2+}$, NH$_4^+$, Na$^+$ etc.) in the leaching solution. The exchange reaction of Mg$^{2+}$, NH$_4^+$, and rare earth ions can be expressed as follows:

$$[Al_2Si_2O_5(OH)_4]_m \cdot 2nRE^{3+}(s) + 3nMg^{2+}(aq) \rightleftharpoons [Al_2Si_2O_5(OH)_4]_m \cdot [Mg^{2+}]_{3n}(s) + 2nRE^{3+}(aq) \quad (2)$$

$$[Al_2Si_2O_5(OH)_4]_m \cdot 2nRE^{3+}(s) + 3nNH_4^+(aq) \rightleftharpoons [Al_2Si_2O_5(OH)_4]_m \cdot [NH_4^+]_{3n}(s) + nRE^{3+}(aq) \quad (3)$$

When ion-absorbed rare earth ore and the leaching solution come into contact with each other, the cations on the surface of the clay mineral will attract the anions in the leaching solution and make the anions form a liquid film in the leaching solution.The liquid film and the nearby area form the "electric double layer". Studies have shown that the lower cation valence in the pore water of the soil, the thicker the electric double layer formed, and asmaller ion concentration also hasthe same effect. According to the theory of water film [26], anincrease inthe thickness of the electric double layer will greatly change the pores of the soil body, so that the pore water pressure $u_w$ in the soil body increases, the matrix suction $(u_a - u_w)$ decreases, the soil volume water content decreases, and the water-holding capacity of the soil body is weakened.

Since pure water cannot undergo an ion-exchange reaction with clay minerals in ion-absorbed rare earth, the seepage effect of pure water influencesrelatively little change on the pore structure of the soil.The number of large pores inside the sample is small and the connectivity is poor, the air-entry value is relatively high, and the drying rate is small, so the water-holding capacity of the sample is relatively strong. The reaction of an ammonium sulfate solution and clay minerals is more violent than a magnesium sulfate solution; a larger number of large pores were produced and connectivity was enhanced, pore size distribution is more uniform; therefore, the sample dehydrated at a faster rateunderrelatively lowsuction conditions, so the air-entry value of the sample is smaller andthe drying rate is larger. At the same time, Mg$^{2+}$ and NH$_4^+$ undergo an ion-exchange reaction with clay minerals, and the electric double layer formed by the monovalent NH$_4^+$ is thicker than that formed by the divalent Mg$^{2+}$, where itis easier to block the pores under the corresponding conditions, so that the pore water pressure of the soil increases; therefore, the sample is more likely to lose water, and the water-holding capacity is reduced. Therefore, after the leaching of pure water, the 3% magnesium sulfate solution, and the 3% ammonium sulfate solution, the water content reduction rate of the three samples gradually

accelerates in thesoil-water characteristic curve transformationstage, and the water-holding capacity is weakened.

### 3.2. Effect of Leaching Concentration On Water-Holding Characteristics

The soil-water characteristic curves of ion-absorbed rare earth after the leaching of different concentrations of leaching solution are shown in Figure 5. It can be found that the SWCC change law after the leaching of different concentrations of leaching solution is similar, and with the increase inmatrix suction, the volume water content first decreases slowly, then rapidly decreases, and finally tends to be stable and close to 0.

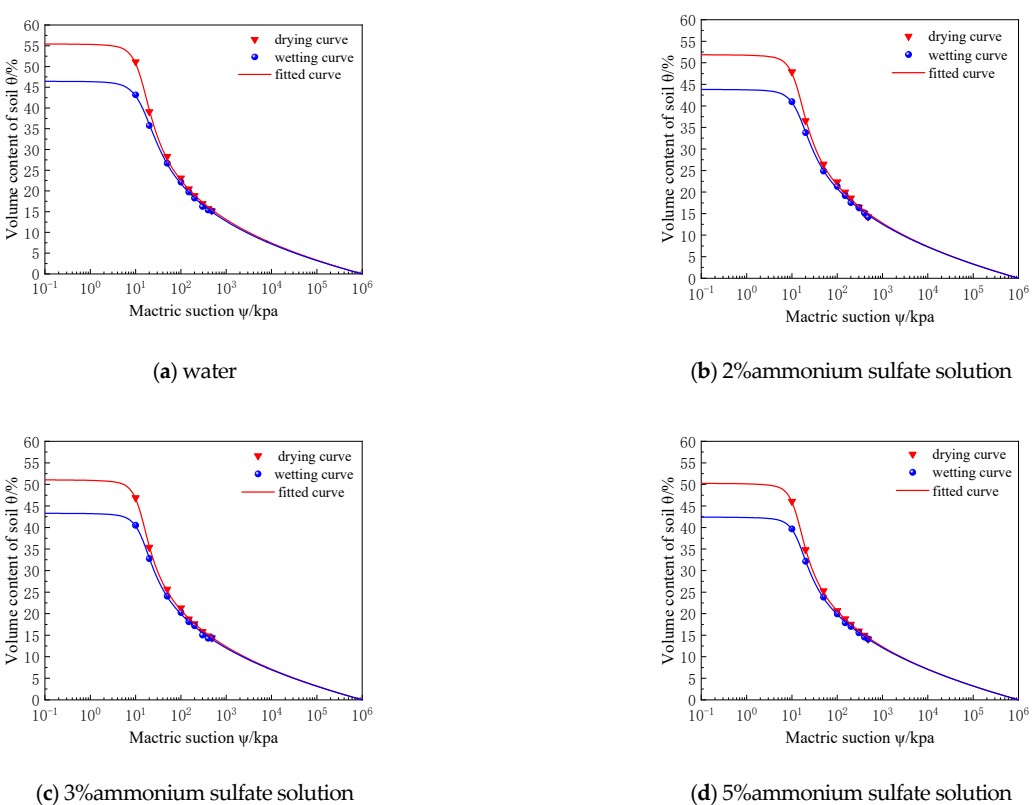

(**a**) water

(**b**) 2%ammonium sulfate solution

(**c**) 3%ammonium sulfate solution

(**d**) 5%ammonium sulfate solution

**Figure 5.** Soil-water characteristic curve under different concentrations of leaching solutions.

The drying and wetting curves of ion-absorbed rare earthafter the leaching of different concentrations of leaching solution are plotted in Figure 6. It can be seen that under the leaching of different concentrations of the leaching solutions, with the increase inthe concentration of the leaching solutions, the drying and wetting curve of ion-absorbed rare earth show a downward shift, and the shift between pure water and 2% ammonium sulfate solution is the most obvious. With the increase inmatrix suction, the water content showed a gentle decrease at first, then rapidly decreased, and, finally, the water content reduction rate gradually decreased, and when the matrix suction was closer to $10^6$ kPa, the drying and wetting curve tended to coincide. When the matrix suction is constant, with the increase inconcentration of the leaching solution, the corresponding water content gradually decreases, that is, the less water stored inside the soil, indicating that the water-holding capacity of the soil is weakened.

(**a**) Drying curve                    (**b**) Wetting curve

**Figure 6.** Drying and wetting curve under different concentrations of leaching solution.

Similar to the mechanism action of different types of leaching solutions on ion-absorbed rare earth, the ion-exchange reaction is the main factor causing the change of pore structure inside the soil [27]. For different concentrations of ammonium sulfate solution, the solute is monovalent $NH^{4+}$, with the increase inion concentration, the thickness of the electric double layer formed decreases, it is less likely to cause pore blockage, the communication effect between the pores inside the soil is relatively better, and the pore size distribution is more uniform. When the concentration of the leaching solution increases, the ion-exchange reaction between the leaching solution and the clay minerals is more violent, and the more obvious the change of the internal pore structure of the sample. With the increase in internal porosity of the soil and pore radius, the greaternumberof large pores and medium pores in the soil form and the better the communication between the pores, as well as the reduction inthe constraining force to the water in the pores; therefore, the sample begins to lose water and the rate is faster under the smaller matrix suction. With the increase inthe concentration of the leaching solution, the air-entry value of the soil gradually decreases, the drying rate gradually increases, and the water-holding capacity is weakened.

## 4. Analysis of "Hysteresis Effect"onSoil-Water Characteristic Curve

*4.1. Effect of Different Leaching Action on the "Hysteresis Effect" of Soil-Water Characteristic Curve*

The drying curve of the ion-absorbed rare earth soil-water characteristic curve under different leaching action is always above the wetting curve, and do not coincide, there is an obvious "hysteresis effect". In order to better quantify the influence of different leaching action on the "hysteresis effect", consider the difference between the drying and wetting curve and the area enclosed by the horizontal and vertical axes to characterize the saliency of the "hysteresis effect" [28], as shown in Equation (4),

$$S_{\text{hysteresis}} = \int_{0.1}^{10^6} (\theta_w)_{\text{drying}} \, d\psi - \int_{0.1}^{10^6} (\theta_w)_{\text{wetting}} \, d\psi \tag{4}$$

where $S_{\text{hysteresis}}$denotesthe saliency of the hysteresis effect, and $\int_{0.1}^{10^6}$ denotes the integration from 0.1 to $10^6$. $(\theta_w)_{\text{drying}}$denotes the drying curve, $(\theta_w)_{\text{wetting}}$denotes the wetting curve, and $\psi$is the matrix suction. Since the matrix suction $\psi$is represented in kPa and the water content is represented in "%", the units in Equation (4) are defined as kPa.

The area enclosed by the drying curve, wetting curve, and the horizontal and vertical axes under different leaching actionsarecalculated by Origin software, and the calculation results are shown in Table 2.

**Table 2.** Area enclosed by the drying–wetting curve and the axis.

| Array | Different Types of Leaching Solutions | | | Different Concentrations of Leaching Solutions | | | |
|---|---|---|---|---|---|---|---|
| | Water | 3% Magnesium Sulfate | 3% Ammonium Sulfate | Water | 2% Ammonium Sulfate | 3% Ammonium Sulfate | 5% Ammonium Sulfate |
| $S_{drying}$/(kPa) | 26, 285, 881 | 24, 846, 699 | 24, 147, 965 | 26, 285, 881 | 24, 508, 822 | 24, 147, 965 | 23, 637, 732 |
| $S_{wetting}$/(kPa) | 21, 827, 473 | 20, 936, 176 | 20, 284, 435 | 21, 827, 473 | 20, 486, 381 | 20, 284, 435 | 19, 812, 978 |
| $S_{hysteresis}$/(kPa) | 4, 458, 408 | 3, 910, 523 | 3, 863, 530 | 4, 458, 408 | 4, 022, 441 | 3, 863, 530 | 3, 824, 754 |

The saliency of the "hysteresis effect" of the soil-water characteristic curves after the leaching action of different types of leaching solution is shown in Figure 7. It can be found that the "hysteresis effect" of the sample afterthe leaching of pure water is the most conspicuous, while it is significantly reduced after the leaching of the 3% magnesium sulfate solution and the 3% ammonium sulfate solution, withthe former being slightly larger than the latter. The analysis believes that the significance of the "hysteresis effect" is greatly related to the pore structure change caused by the chemical reaction between the leaching solution and rare earth mineral particles. Pure water can not undergo ion-exchange reaction with rare earth elements in ion-absorbed rare earth, the change of surface pores of clay mineral particles is small during leaching, and the wetness of clay particles is lower in the process of wetting, so the contact angle formed is larger.According to Equation (4), it can be seen that the matrix suction is smaller at the same water content under the leaching of pure water, so the "hysteresis effect" is more significant. Compared with pure water, $Mg^{2+}$ and $NH_4^+$both can undergo an ion-exchange reaction with rare earth elements, the internal pore skeleton structure of the sample changes greatly during leaching, the ion-exchange reaction makes the clay mineral particles more uniform, the fine "throat" connecting the pores expands or disappears.During the drying and wetting process, "throat" has reduced the constraining force on water, the number of medium pores and large pores in the sample increases, and it is easier to lose and absorb water under the same matrix suction; the saliency of the "hysteresis effect" is obviously reduced.

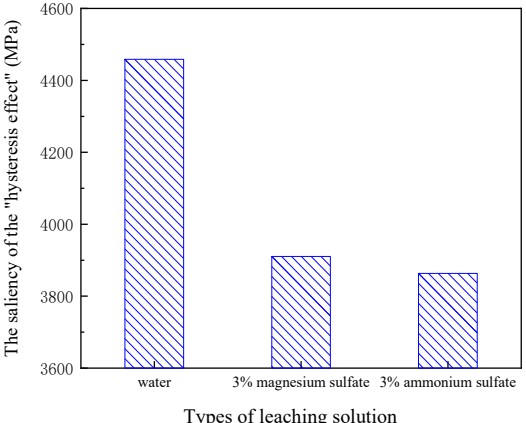

**Figure 7.** The saliency of the "hysteresis effect" of the soil-water characteristic curve under different types of leaching solution.

It can be seen from Table 2 that. with the increase inthe concentration of the leaching solution, the saliency of the "hysteresis effect" of the soil-water characteristic curve gradually decreases, and it decreases most obviously from pure water to the 2% ammonium sulfate solution. Additionally, the saliency of the "hysteresis effect" after different concentrations of leaching solution is shown in Figure 8.

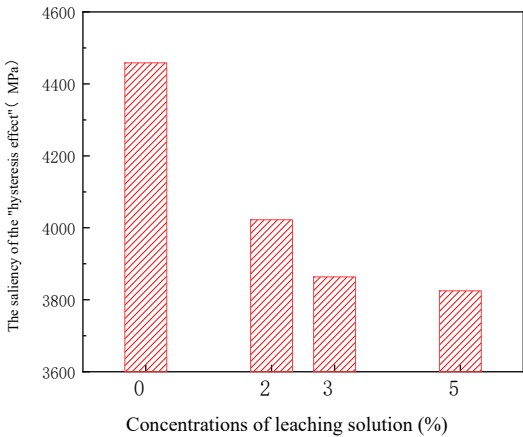

**Figure 8.** The saliency of the "hysteresis effect" of the soil-water characteristic curve under different concentrations of leaching solutions.

It can be found that the relationship between the concentration of the leaching solution and the saliency of the "hysteresis effect" of the soil-water characteristic curve is well fitted by using the exponential function, and the $R^2$ =0.991. The relationship between the two is as follows.

$$S_{hysteresis}=3766.93+693.6\times0.58^x \tag{5}$$

where $S_{hysteresis}$ denotes the saliency of the hysteresis effect, MPa; $x$ is the concentration of the leaching solution, "%".

Under the sameleaching time, with the increase inthe concentration of the leaching solution, the more intense the ion-exchange reaction between the sample and the leaching solution, the more fully the micro-fractures on the surface of clay mineral particles develop, the bigger amounts of medium pores and large pores in the sample form, the smallerthe constraining force of the small "throat" connecting the pores on waterbecome.Consequently, the sample is more likely to lose and absorb water under the same matrix suction, the saliency of the "hysteresis effect" of the soil-water characteristic curve shows a decreasing trend. At the same time, from pure water to the 2% ammonium sulfate solution, the saliency of the "hysteresis effect" decreased the fastest, and the decrease rate gradually slowed down as the concentration increased. The main reason is that the ion-exchange reaction and seepage rearrange the soil particles, leached from pure water to the 2% ammonium sulfate solution, the change inthe particle skeleton is the most obvious, and with the increase inthe concentration, the particle skeleton tends to be stable, therefore the decrease rate of "hysteresis effect" gradually reduced.

### 4.2. Mechanism Analysis of "Hysteresis Effect"

The dryingcurve obtained by soil dehydration does not coincide with the wetting curve obtained by soil-water absorption, and there is an obvious "hysteresis" phenomenon. There are many factors affecting the "hysteresis effect" of the soil-water characteristic curve, and there are, mainly, the following two reasons for the analysis.

(1)   The contact angle between the soil and the liquid.

When the liquid and the solid are in contact with each other, if the interaction force (i.e., adhesion) between the solid surface molecules and the fluid molecules is relatively greater than the internal interaction force of the liquid molecule (i.e., cohesion), the wetting phenomenon will occur between the two, conversely, it will not occur. The angle formed between the tangent line of the liquid surface and the solid surface, the contact angle $\alpha$, can be used to characterize the wetness of the liquid to the solid.

Due to the heterogeneity of the surface of soil particles, when there is pore water in the pores of the soil, the presence of the hysteresis effect leads to the contact angle is not the only one, its value is generally between the forward contact angle and the backward contact angle, and the backward contact angle during drying is significantly smaller than the forward contact angle during wetting, as shown in Figure 9.

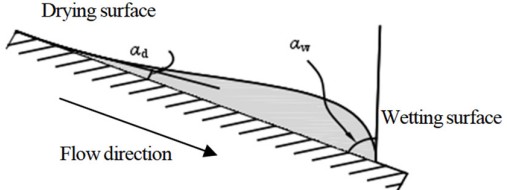

**Figure 9.** Schematic diagram of the contact angle between the wetting surface and the drying surface.

According to the capillarity theory, when considering the contact angle action, the matrix suction can be expressed by Equation (6).

$$u_a - u_w = \frac{2T_s \cos\alpha}{R_s}$$ (6)

where $(u_a - u_w)$ is the matrix suction, $T_s$ is the surface tension of the water, $\alpha$ is the size of the contact angle, and $R_s$ is the radius of curvature of the bending surface.

According to the theory of water film, when the contact angle is small, the corresponding surface tension is greater. In the process of drying and wetting, the contact angle during drying is smaller than that when wetting, so its surface tension is larger, $\cos\alpha$ is also larger. Therefore, when the water content is constant, the matrix suction corresponding to the drying process is greater than that of the wettingprocess, resulting in the "hysteresis effect".

(2)   The "ink-bottle" effect.

The size of the pores inside the soil is different, and different pores are connected to each other through small throat. Due to the size difference between the pores and the throat, during the drying and wetting, the flow of water into or out of the pores must be "constrained" by the smaller throat, the so-called "ink-bottle" effect, which can be explained by capillary action. Assume that the radius of the "throat" with a relatively small size is r and the radius of the "pore" with a relatively large size is R, as shown in Figure 10.

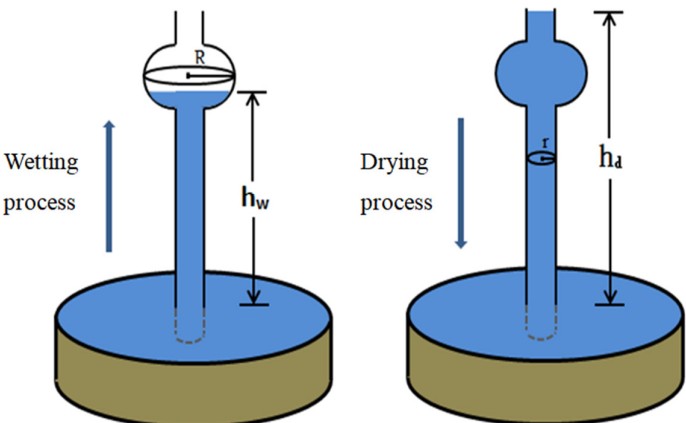

**Figure 10.** Schematic diagram of the mechanism of action of the "ink-bottle" effect.

In the wetting process, water flows into the pores, and the highest position that can be reached by the capillary action is controlled by a larger radius R, which corresponds to the matrix suction $u_a - u_w = 2T_s / R$. For the drying process, assuming that the pores and "throat" are filled with water beforehand, the maximum height of capillary action is $h_d$ ($h_d > h_w$), and when the drainage process reaches equilibrium, the matrix suction is controlled by a smaller radius r, and the matrix suction at this time is $u_a - u_w = 2T_s / r$. Due to the R>r, the matrix suction during drying is greater than that of the wetting process.

## 5. Conclusions

Under different leaching actions, with the increase inmatrix suction, the water content showed the law of "first slowly decreased, then rapidly decreased, and finally stabilized", when the matrix suction was closer to $10^6$ kPa, the drying and wetting curves tended to coincide.This is consistent with the conclusions of other existing leachingtest studies based on different methods to obtain soil-water characteristic curves.

For the different types of leaching solution, the water-holding capacity of the soil ranged from strong to weak forpure water, the 3% magnesium sulfate solution, and the 3% ammonium sulfate solution. With the increase inconcentration, the water-holding capacity of the soil gradually decreased, and decreased most significantly when the concentration vary from 0 to 2%, according to water film theory, the increase inthickness of the electric double layer leads to the same change inpore water pressure and reduction inmatric suction. Consequently, the water-holding capacity of rare earth ores is closely related to the ionic valence and type of solutes in the leaching solution.

The salience of the "hysteresis effect" of soil-water characteristic curves after the leaching of different types of leaching solution was pure water, the 3% magnesium sulfate solution, and the 3% ammonium sulfate solution. Additionally, the salience of the "hysteresis effect" of the soil-water characteristic curve decreases sequentially with the increase inthe concentration. The soil-water characteristic curve produces the "hysteresis effect", mainly due to the contact angle effect and the ink-bottle effect.

**Author Contributions:** Conceptualization, Z.G. and L.L.; methodology, Z.G.; software, K.Z.; validation, Z.G., L.L. and K.Z.; data curation, K.Z.; writing—original draft preparation, Z.G.; writing—review and editing, Z.G.; visualization, L.L and T.T.; supervision, Z.G.; project administration, Z.G.; funding acquisition, Z.G. and X.W. All authors have read and agreed to the published version of the manuscript.

**Funding:** This research was funded by the National Natural Science Foundation of China (52004106, 52174113); the Natural Science Foundation of Jiangxi Province (20224BAB214035).

**Data Availability Statement:** Not applicable.

**Acknowledgments:** Thanks for the great effort by editors and reviewers.

**Conflicts of Interest:** The authors declare no conflicts of interest.

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
