# Peer review of "Influence of Leaching Solution on the Soil-Water Characteristics of Ion-Absorbed Rare Earth Minerals and Its Hysteresis Effect"

_minerals, doi:10.3390/min13050637_

Round 1

Reviewer 1 Report

All suggestions and comments are in Word file

There are many grammatical mistakes. Please check the manuscript for grammar and English carefully.

Author Response

Dear Reviewers:

Thank you for your comments and suggestions concerning our manuscript entitled “Influence of leaching solution on the soil-water characteristics of ion-absorbed rare earth minerals and its hysteresis effect” (ID: 2331558). Those comments are all valuable and very helpful for revising and improving our paper, as well as the important guiding significance to our researches. We have studied comments carefully and have made correction which we hope meet with approval. Revised portion are marked in red in the paper. The main corrections in the paper and the responds to the reviewer’s comments are as flowing:

  1. Response to comment: There are many grammatical mistakes. Please check the manuscript for grammar and English carefully.

Response: We have checked and corrected the grammar and English carefully throughout all the manuscript to ensure the correctness and rigor of the language.

  1. Response to comment: L11: space needed for Fredlund& and in all the manuscript

Response: As suggested by the reviewer, we have modified the required spaces here throughout all the manuscript.

  1. Response to comment: The introduction section: the novelty of the study was not presented clearly. Authors should point out the novelty of their study clearly at the last paragraph of introduction.

Response: As requested by the reviewer, we have added statement in “Introduction” to point out the novelty of this study. The added statement is marked in redat the last paragraph of introduction.

  1. Response to comment: L64: Space needed for LongnanZudongRare

Response: As suggested by the reviewer we've modified the required spaces here throughout the manuscript.

  1. Response to comment: The literature survey in the introduction section can be enrich by adding some recent references related to leaching of rare earth, such as: Nanomaterials 2022, 12(13), 2305 doi:10.3390/nano12132305; Materials 2022, 15(3), 1211 doi:10.3390/ma15031211

Response: As suggested by the reviewer, we have enriched this section by adding the latest references related to ionic rare earth leaching, The added article is marked in the References section.

  1. Response to comment: The experimental section: L74: Space needed for samples are

Response: As suggested by the reviewer we've modified the required spaces here throughout the manuscript.

  1. Response to comment: L75: Space needed for Province, China. The

Response: As suggested by the reviewer we've modified the required spaces here throughout the manuscript.

  1. Response to comment: What are the solid to liquid ratio used and temperature during the leaching process? Please state into experimental section.

Response: As requested by the reviewer we have added the relevant explanations of liquid-solid ratio and temperature to the experimental section, according to relevant studies, the optimal liquid-solid ratio of in-situ leaching is 0.6:1, so the liquid-solid ratio of our experiment was set to 0.6:1, and the temperature was set to 25°.The added explanation is marked in the corresponding section.

  1. Response to comment: How many times the experiments were performed?

Response: In order to avoid the accident and error of the experiment and ensure the objectivity of the experiment results, we carried out three sets of repeatability experiments for each group of working conditions to obtain more accurate experimental data.

  1. Response to comment: Change hours to h.

Response: As suggested by the reviewer we've modified the expression here throughout the manuscript.

  1. Response to comment: Results and Discussion section: Fig 3−8 error bars are missing.

Response: As pointed out by the reviewers, since the experiment data in this study were obtained by the soil-water characteristic curve pressure plate apparatus, under the same experiment conditions, the wetting and drying data of the obtained specimens were consistent, and after repeated experiments for each group of the same working conditions, the standard deviation of the data changed very little, so the error bars were not represented in the figure. In view of the comments of the reviewers, we will add error bars to the data charts that require obvious changes in standard deviation in subsequent experimental studies to achieve a more accurate and objective analysis.

  1. Response to comment: The conclusion section: the numbering list should be removed and replaced by paragraphs.

Response: As suggested by the reviewer, we've removed the numbering list of the conclusion section and replace by paragraphs.

Reviewer 2 Report

According to the text of the article, it is necessary to correct the numbering of equations and tables. There is a discrepancy.

Author Response

  1. Comment: According to the text of the article, it is necessary to correct the numbering of equations and tables. There is a discrepancy.

Response: As suggested by the reviewer, we have corrected the equations and table numbers in the article to ensure consistency between the presentation and the corresponding charts.

Reviewer 3 Report

The paper, presents interesting ideas, on a matter of clear scientific and industrial interest, but, on the other hand it’s not so innovative in the approach and in the contents.

The main contribution of the paper is the analysis of "hysteresis effect" that is well described, but some other parts of the paper have to be improved.

So, I recommend that this paper be accepted after minor revision:

1. “Intoduction” paragraph have to be improved with a wider view on ion absorbtion, so also references can be improved.

2. The importance of results obtained have to be better explained in the contest of rare earths scenario. So, also the conclusions may be more exhaustive by a comparative analysis with othet technology.

3. English language have to be improoved. 

The paper, presents interesting ideas, on a matter of clear scientific and industrial interest, but, on the other hand it’s not so innovative in the approach and in the contents.

 The main contribution of the paper is the analysis of "hysteresis effect" that is well described, but some other parts of the paper have to be improved.

So, I recommend that this paper be accepted after minor revision:

Author Response

  1. Comment: “Introduction” paragraph has to be improved with a wider view on ion absorption, so also references can be improved.

Response: As suggested by the reviewer, we have made more complete additions to the research progress of ion-adsorbed rare earths in this paper, and have also adjusted in the reference section. The added article is marked in red at the References section.

  1. Comment: The importance of results obtained have to be better explained in the contest of rare earths scenario. So, also the conclusions may be more exhaustive by a comparative analysis with other technology.

Response: As suggested by the reviewer,on the basis of comparing and analyzing the results of other experimental studies, we have made a more comprehensive explanation of the significance and impact of the experimental conclusions, and once again highlighted the important scientific significance of this experimental research.The added expression is marked in red at the conclusion section.

  1. Comment: English language have to be improved.

Response: We have checked and corrected the grammar and English carefully throughout all the manuscript to ensure the correctness and rigor of the language.

Round 2

Reviewer 1 Report

Accept in present form

Minor editing of English language is required